# Airborne Pheromone Quantification in Treated Vineyards with Different Mating Disruption Dispensers against *Lobesia botrana*

**DOI:** 10.3390/insects11050289

**Published:** 2020-05-09

**Authors:** Aitor Gavara, Sandra Vacas, Ismael Navarro, Jaime Primo, Vicente Navarro-Llopis

**Affiliations:** 1Centro de Ecología Química Agrícola, Instituto Agroforestal del Mediterráneo, Universitat Politècnica de València, Edificio 6C, Camino de Vera s/n, 46022 Valencia, Spain; jprimo@ceqa.upv.es; 2Ecología y Protección Agrícola SL, Pol. Ind. Ciutat de Carlet, 46240 Valencia, Spain; ismael.navarro@epa-ecologia.com

**Keywords:** European grapevine moth, Tortricidae, passive dispensers, aerosol devices, air samples, GC–MS/MS, pest control, field trials

## Abstract

Mating disruption (MD) is widely used against the European grapevine moth (EGVM), *Lobesia botrana* (Denis and Schiffermüller; Lepidoptera: Tortricidae), by installing passive dispensers or aerosol devices. The present work reports a new sampling and quantification methodology to obtain absolute data about field airborne pheromone concentration based on air samplings and sensitive chromatographic-spectroscopic methods. Samplings were performed in fields treated with passive dispensers or aerosol devices at different moments throughout the crop cycle to study how they act and how the disruption is triggered. Moreover, pheromone adsorption and releasing capacity of vine leaves were studied to elucidate their role in the disruption. Although both types of dispensers were effective in limiting the damage inflicted by EGVM, they performed differently and provided different airborne pheromone concentration profiles. Results also proved that leaves were able to adsorb and release part of the airborne pheromone acting as subsequent and additional pheromone sources. This fact could explain the different concentration profiles. Moreover, our results suggest that lower pheromone emission than that of the current passive dispensers still could provide an adequate performance in the field. Competitive mechanisms involved in MD using both dispensers, the dynamics of the airborne pheromone throughout the time and the importance of the canopy are discussed.

## 1. Introduction

Mating disruption (MD) is widely used as an environmentally friendly pest control method in economically important crops due to its high effectiveness, specificity and the lack of toxic residues on fruit, in contrast to the conventional methods based on the use of pesticides [1,2]. Nevertheless, the higher cost of mating disruption compared with the traditional control in some crops is sometimes detrimental for its implementation [3,4,5,6]. For this reason, the studies focused on the optimization of the pheromone used, quantity and releasing mode are essential to reduce the implementation costs of this technique and make it more available to growers. This fact is of great importance to introduce MD to control those species for which the cost of implementation is too high to be assumed by the growers and to optimize the use of pheromone for those that are already controlled by MD.

Once mating disruption is proven effective against a pest, the pheromone cost is crucial for the implementation of this technique. The final cost of mating disruption depends on pheromone production costs [7,8], which are related to: (1) chemical synthesis and economies of scale, (2) the quantity of pheromone needed and (3) the pheromone emission rate and the number of dispensers required per hectare. All these parameters are related to the biology of the target pest and the mechanisms that trigger the mating disruption. In general, mating disruption may be governed by different mechanisms acting almost simultaneously but there is usually a principal mechanism responsible for the disruption. These mechanisms can be classified as competitive or non-competitive. In the case of competitive disruption, males, females, and their signals have no impairments, therefore, females compete with the artificial pheromone sources (dispensers) and the efficacy of the treatment will depend mainly on pest density. On the other hand, when non-competitive disruption is acting, the signal, males or females are impaired since the beginning of sexual activity, making the efficacy pest-density-independent [9].

Hand applied passive dispensers are traditionally the most commonly employed devices to implement mating disruption as a pest control technique [1,2,10,11]. They consist of little containers of plastic materials filled or impregnated with the pheromone, which is released passively through their walls. These dispensers are deployed at high densities of more than 250 units/ha [9,12,13]. Using this type of dispenser, competitive attraction seems to be the main mechanism acting in disrupted fields [14]. Alternatively, automatic aerosol devices are being increasingly used with good efficacy against many moth pests [2,15,16,17]. They are mechanical devices connected to a pressurized metal can, which is loaded with an appropriate formulation of the pheromone, a diluent, and a propellant, usually an inert gas. These devices can be programmed for releasing equal amounts of the formulation containing the pheromone via emissions at regular time intervals when moths are active. Unlike passive dispensers, these devices are deployed at very low densities (2–5 units/ha) due to their high pheromone release [9]. Non-competitive disruption triggered by desensitization of males along their wide plumes was an explanation of their performance [7], although a competitive mechanism cannot be dismissed and most current studies report that males are drawn away from females towards the pheromone sources in a mechanism called “induced allopatry” [9,16,18].

The European grapevine moth (EGVM), *Lobesia botrana* (Denis and Schiffermüller; Lepidoptera: Tortricidae), is a key pest in vineyards, which has been traditionally controlled using mating disruption [19,20] by employing hand applied passive dispensers at high densities (500 or, more recently, 250 units on average per hectare) providing a continuous emission of (*E,Z*)-7,9-dodecadien-1-yl acetate [12,13,21,22]. Currently, growers can disrupt EGVM using either passive pheromone dispensers or aerosol devices to apply mating disruption against EGVM. In this pest, the efficacy of aerosols is scarcely reported in the scientific literature but with promising results [2,23]. In addition, these studies highlighted three inherent advantages of this technique: the lower number of dispensers per hectare needed, the reduction of labor costs and the reduction of plastic disposal compared with passive dispensers.

Despite the effectiveness reported for both types of dispensers, the reduction of their cost and improvement of their efficacy are still necessary to increase the use of mating disruption. For this reason, better knowledge of the mode of action of each one must be accurately studied. Most of the studies performed on different pests are based on empirical and indirect measurements of the efficacy based on parameters such as fruit injury levels, dispenser load, distribution, etc. [5,18,24,25,26,27]. The quantification of airborne pheromone concentrations present in the field, the study of pheromone distribution from the time of their releasing and the interaction with the canopy could offer an interesting approach with valuable information to improve the efficacy of pheromone-releasing systems [28].

Unfortunately, there is scarce information regarding airborne pheromone concentrations due to the difficulty of its quantification. Three techniques were described in the past to measure airborne pheromone [29]: (1) field electroantennography (EAG) measurements, (2) single sensillum recordings and (3) air sampling by collecting airborne volatiles on filters and their chemical analysis. Field EAG offers more instantaneous and quick measurements that allow studying temporal and spatial pheromone fluctuations. However, data obtained with these methods could be misinterpreted due to the interaction of pheromone and host–plant detection, differences between individual antennae and the lack of linearity of the detector [29]. Moreover, this procedure needs precise calibrations due to the complexity of the environmental matrix and insect fitness, and it also depends on a proper handling of insect preparations. Single sensillum recordings have the advantage of the elimination of other semiochemicals effect in the signal. However, it is more difficult to carry out than field EAG due to the fragile and laborious preparation of the antennae, besides not being effective for the interpretation of the signal when a pheromone blend triggers an insect response. There are works that attempted to study airborne pheromone concentration in fields treated against EGVM using field EAG in the 1990s [29,30,31,32]. In these experiments, they were capable to obtain relative data to see differences in airborne pheromone concentration by studying different parameters such as different seasons (spring and summer), height of dispensers deployment, their density, presence of developed foliage, distance from the center of the plot, etc., but they had several difficulties due to the field EAG drawbacks.

Air sampling followed by chemical analysis of collected samples for pheromone quantification is a technically feasible method that can provide the most accurate and absolute data about airborne pheromone concentration in treated fields. However, it provides time-averaged data because relatively long sampling periods are needed to detect the minute quantities of pheromone collected [29]. The low detection level thresholds required were, in the past, the main difficulty for these measurements, but currently, due to the lower limits of detection achieved by analytical instruments, an accurate quantification of airborne substances is possible. Many methods have been developed to detect environmental and health-damaging substances at very low concentrations, i.e., dioxins and furans are quantified in air at concentrations of femtograms (10^−15^ g) per cubic meter [33,34,35]. Airborne psychotropic substances were quantified at nanograms (10^−9^ g) per cubic meter in different Italian cities [36] as an example of substances at similar concentration levels to airborne pheromones. These quantifications can be achieved with high volume collectors, effective adsorbents, and more sensitive chromatographic and spectroscopic methods such as gas chromatography coupled with triple quadrupole mass spectrometers (GC/MS/MS). Although there are many protocols for quantifying environmental and health damaging substances, the similarities of pheromone chemical structures with organic compounds present in the environmental matrix and the lack of characteristic functional groups still make their identification and quantification more difficult.

In this work we have developed a method that allowed us to successfully quantify airborne concentrations of EGVM pheromone in vineyards treated with two commercial releasing systems: passive dispensers and aerosol devices. These quantifications have been performed in three different moments throughout the vineyards crop cycle for both types of dispensers, while comparing their efficacy. In addition, foliar samples of these treated vineyards were taken to analyze their pheromone content and check their role as subsequent and additional pheromone sources. All the data collected has been employed to discuss the operating mode of both dispensers and the competitive mechanisms governing the mating disruption of the EGVM with each pheromone emission system.

## 2. Materials and Methods

### 2.1. Field Locations and Mating Disruption Systems

Studies were conducted between March and October 2018 in vineyards for winery trained onto trellises. The experiment was a complete block design with two locations (Location 1 and Location 2) and three treatments within each site. Location 1 was located in Venta del Moro (Valencia, Spain; UTMs 39.482608, −1.318822) with a plantation pattern of 2.5 m × 1.5 m (cultivar Bobal) and Location 2 in Moixent (Valencia, Spain; UTMs 38.808779, −0.815917), also with a plantation pattern of 2.5 m × 1.5 m (cultivar Monastrell). The three different treatments included at each location were: (1) Isonet L (MD treatment with the passive dispensers Isonet^®^ L (Shin-Etsu Chemical Co., Tokyo, Japan) at a density of 500 dispensers/ha); (2) Checkmate (MD treatment with the aerosol devices CheckMate^®^Puffer^®^ LB (Suterra LLC, Bend, OR, USA) at a density of 3 dispensers/ha) and (3) untreated control (UTC), without any type of treatment against EGVM. None of the fields of either location received any additional insecticide or fungicide application. The three treatments were applied at each location by delimiting 2 ha regular areas. The UTC areas were located 500 m upwind from the MD treatments at each site. The aerosol devices were installed following the manufacturer instructions and programmed to release the pheromone via emissions each quarter hour from 18:00 h to 06:00 h.

Male flight was monitored weekly at both locations from 21 March 2018 and the MD treatments were not deployed until the first catch was detected at any location, which was near the end of April. Then, the dispensers were set on 24 April 2018 at Location 1 and on 23 April 2018 at Location 2.

### 2.2. Airborne Pheromone Quantification

#### 2.2.1. Chemicals and Reagents

All solvents and chemical reagents used (HPLC and Synthesis grade, respectively) were purchased from Merck (Darmstadt, Germany). All reactions were carried out under a nitrogen atmosphere with dry solvents, unless otherwise noted. Solvents were dried under standard procedures. All purchased chemicals were used without further purification. Macherey-Nagel Silica gel 60 F254 TLC plates (Fisher Scientific SL, Madrid, Spain) were used for analytical thin layer chromatography (TLC) technique. Plates were visualized by using a UV lamp or using an appropriate stainer (*p*-anisaldehyde in ethanol/aqueous H_2_SO_4_/CH_3_CO_2_H or 10% solution of phosphomolybdic acid in ethanol and heat as developing agents).

(*E*,*Z*/*E*)-7,9-dodecadien-1-yl acetate was supplied by Ecología y Protección Agrícola S.L. (Carlet, Valencia, Spain). A high purity standard of (*E*,*Z*)-7,9-dodecadien-1-yl acetate (ca. 95%) was obtained after purification of the supplied sample by gravity column (silica gel containing 10% of AgNO_3_; eluent: 3% Et_2_O/hexane), which had spectroscopic properties identical to those described in the literature [37].

Due to the abundance of pheromone structurally related compounds in air collected samples, a straight chain fluorinated hydrocarbon ester (heptyl 4,4,5,5,6,6,7,7,8,8,9,9,9-tridecafluorononanoate), was selected as an internal standard in order to improve both sensitivity and selectivity for MS/MS method optimization. See the next section for experimental details

#### 2.2.2. Synthesis of Heptyl 4,4,5,5,6,6,7,7,8,8,9,9,9-Tridecafluorononanoate (TFN)

To a solution of 4-dimethylaminopyridine (DMAP; 15 mg) in dichloromethane (DCM; 30 mL), 4,4,5,5,6,6,7,7,8,8,9,9,9-tridecafluorononanoic acid (500 mg, 1.3 mmol) was added. After stirring during 60 min at room temperature, 1-heptanol (0.21 mL, 1.5 mmol) was added and the mixture was refluxed for 24 h. After this period, the solvent was removed under vacuum and the crude was purified by column chromatography (silica gel; eluent: 1% Et_2_O/Hexane) to yield heptyl 4,4,5,5,6,6,7,7,8,8,9,9,9-tridecafluorononanoate (281 mg, 45%) as a colorless oil. MS (70 eV, *m/z*): 393 (10%), 375 (40%), 373 (5%), 132 (10%), 98 (30%), 83 (15%), 70 (100%), 69 (70%), 57 (90%) and 56 (90%).

#### 2.2.3. Field Air Sampling

Air sampling was carried out with a high-volume air sampler (CAV-A/Mb, MCV, Barcelona, Spain) provided with an adsorbent sandwich composed by alternate layers of 20 g/L polyurethane foam (MCV, Barcelona, Spain) and adsorbent resin (XAD-2, Sigma-Aldrich, Madrid, Spain). The sandwich structure was designed according to literature [38,39] and preliminary laboratory experiments.

The air samples of the Isonet treatment were taken at Location 1, whilst the air samples of the Checkmate treatment were taken at Location 2 due to the availability of electrical supply for the air sampler in nearby field warehouses. From that electrical source, an extension cord was connected in order to place the high-volume air sampler in the center of each treated area. Samplings of each treatment were carried out during three different plant growth phenological stages (BBCH scale) according to Lorenz et al. [40] throughout the crop cycle: (1) 4 samples after dispensers setting at BBCH 00-15 (May–June) for both, Checkmate and Isonet treatments; (2) 4 samples for Checkmate treatment and 3 samples for Isonet L treatment at the middle of the crop cycle at BBCH 68-75 (last week of July and first of August) and (3) 3 more samples for each one of both types of dispensers at the end of the crop cycle, before the harvest BBCH (81-89; between 11 and 27 September 2018). Samplings were initiated at 12:00 h and the sampling time was established at 48 h with a flow of 15 m^3^/h to cover two nights because of the aerosol devices activity. Once the samples were collected, the different parts of the sandwich were stored individually in a freezer at 3 °C in airtight bags until their extraction. Due to the long sampling periods (48 h/sampling), we were not able to perform replicates for each treatment and phenological stage at both locations. For example, the first and the last samples for the same phenological stage would be taken about 15 days apart, too far in time to be comparable.

#### 2.2.4. Sample Extraction and Filtration

Foam-resin adsorbent sandwiches were extracted using a 250 mL Soxhlet apparatus, bringing 300 mL of hexane to boil. Temperature was set at 90 °C for 6 h obtaining approximately 15 Soxhlet cycles. Each extract was concentrated up to 5 mL using a rotary evaporator set at 30 °C and finally to 1 mL, blown down with a gentle stream of nitrogen. Then, the samples were cleaned up through a gravity column (0.5 cm × 15 cm) packed, from bottom to top, with a glass wool plug and 500 mg of silica gel pre-eluted with hexane. The column was sequentially eluted with hexane (5 mL) and 2% Et_2_O/hexane (12 mL), collecting three fractions of 5 mL. Fractions two and three containing the pheromone were concentrated up to ca. 1.5 mL using a rotary evaporator set at 30 °C, and finally to 0.5 mL in a 2 mL GC screw-cap (Fisher Scientific SL, Madrid, Spain) blown down with a gentle stream of nitrogen. Occasionally, the column was rinsed with 100% Et_2_O in order to corroborate the complete elution of the pheromone from samples.

#### 2.2.5. Pheromone Quantification

Ten microliters of an internal standard solution of TFN were added to the abovementioned fraction containing the pheromone for the final chemical analysis and quantification. Each extract was then analyzed using a TSQ 8000 Evo triple quadrupole MS/MS instrument operating in SRM (selected reaction monitoring) mode using electron ionization (EI +), coupled with a Thermo Scientific™ TRACE™ 1300 GC. All injections were made onto a ZB-5 (30 m × 0.25 mm × 0.25 mm) fused silica capillary column (Phenomenex Inc., Torrance, CA, USA). The oven was held at 60 °C for 1 min then was raised by 10 °C/min up to 110 °C, maintained for 5 min, raised by 3 °C/min until 150 °C and finally raised by 35 °C/min up to 300 °C held for 5 min. The carrier gas was helium at 1.5 mL/min.

For each target compound—TFN and the main component of EGVM pheromone, (*E*,*Z*)-7,9-dodecadien-1-yl acetate—the MS/MS method was optimized by selecting the precursor ion and the product ions that provided the highest selective and sensitive determinations (Table 1). TFN transition 2 and EGVM transition 3 were the ones employed to obtain the chromatographic areas. In the case of EGVM signal, the other transitions were monitored for confirmatory purposes to have increased selectivity when several peaks appear near to the pheromone peak retention time.

The amount of pheromone and the corresponding chromatographic areas were connected by fitting a linear regression model, *y = a + bx*, where *y* is the ratio between pheromone and TFN areas and *x* is the amount of pheromone.

The percentage of pheromone recovery after the extraction was checked by spiking known pheromone concentrations to a sandwich previously exposed to the same procedure of air sampling in pheromone-free places and, thus, containing environmental matrix. The chemical analysis showed a mean pheromone recovery rate of 92.8%.

### 2.3. Adsorbed Pheromone on Leaves

#### 2.3.1. Adsorption Capacity and Emission Assays

The potential of vine leaves for airborne pheromone adsorption was explored in an experimental plot in Experiment 1 and 2, and later in the commercial field conditions of Location 2 (Experiment 3).

In Experiment 1, a single aerosol emission was made 2 m away from vine leaves—simulating the distance between two rows in a vineyard. Then, 8 g of leaves in the emission influence area were randomly collected immediately (09:00 h) and 4 h, 8 h and 24 h after application. The collected leaf samples were extracted to quantify their pheromone content and to study the release profile. This experiment was replicated 4 times in different points along the vine row (with at least 4 m between emissions) and at different days (2–3 days between emissions).

In Experiment 2, an aerosol installed 2 m away from vine leaves, and 2.5 m from the ground level with the same installation instructions as in field conditions, was programmed to deliver emissions every 15 min from 18:00 h to 06:00 h. Then, 8 g of leaves were collected at three different times: 09:00 h, 13:00 h and 17:00 h (3, 7 and 11 h, respectively, after the last emission), and their pheromone content was quantified. Four replications of this experiment were conducted in previous non-treated areas, at different points along the vine row separated by at least 4 m among the emissions. Moreover, each repetition was done at intervals of 2–3 days.

Once the potential for pheromone adsorption was demonstrated, Experiment 3 was carried out in real field conditions in mid-August at Location 2. Given that passive dispensers are evenly distributed along the fields, leaves were sampled only in the center of this field. However, as the aerosols were deployed along the perimeter of the fields, we decided to sample leaves randomly in the border—along the row where the aerosol was installed—and in the center of the field (70 m from the closest aerosol device) to check differences. Each sample consisted of 8 g of leaves, and three replicates were taken in all cases. All samples were taken at 12:00 h, 6 h after the last puffer emission.

#### 2.3.2. Leaf Sample Extractions

Each 8 g sample was soaked in 300 mL hexane inside a 500 mL round-bottom flask. Each flask was partially immersed in an ultrasonic bath for 30 min and the hexane extract was filtered and concentrated using a rotary evaporator up to 1.5 mL. The solution was transferred to an eppendorf to be centrifuged for 5 min at 3000 rpm. The supernatant was recovered, and to ensure the complete extraction of the pheromone the residue was re-extracted with 1 mL of hexane. Both solutions were concentrated separately, under a gentle nitrogen stream, up to 0.5 mL and first analyzed by gas chromatography using a flame ionization detector (GC/FID) in a Clarus 500 GC (PerkinElmer Inc., Wellesley, MA, USA) using *n*-dodecane as an internal standard. All injections were made onto a ZB-5 (30 m × 0.25 mm × 0.25 µm) column (Phenomenex Inc., Torrance, CA), held at 100 °C for 1 min and then, raised by 25 °C/min up to 250 °C and maintained for 3 min. The carrier gas was helium at 1.5 mL/min. The amount of pheromone and the corresponding chromatographic areas were connected by fitting a linear regression model, *y = a + bx*, where *y* is the ratio between pheromone and *n*-dodecane areas and *x* is the amount of pheromone. The samples that had a pheromone concentration below the quantification limit of the GC/FID were analyzed using a TSQ 8000 Evo triple quadrupole mass spectrometer (GC–MS/MS) coupled with a Thermo Scientific™ TRACE™ 1300 GC. These analyses were performed as described previously in Section 2.2.5. Moreover, the leaves were re-extracted and analyzed to ensure the absence of pheromone in the samples’ residues. Commercial pheromones are usually mixtures of (*E*,*Z*/*E*)-7,9-dodecadien-1-yl acetate. The initial proportion observed between geometrical isomers remained constant along the different measurements, showing no isomerization to the more stable (*E/E*) isomer during this period.

### 2.4. Release Profiles Studies of the Emission Systems

In parallel with the field trials, the pheromone release profiles of the different systems were studied.

Additional passive dispensers were simultaneously aged in nearby areas at Location 1 for approximately 150 days to obtain their release kinetic in a similar way than other studies [41]. Four dispensers were taken from the field each month and their residual pheromone content was extracted and quantified by GC/FID, using the same capillary column and conditions described above in Section 2.3.2. For residual pheromone extraction, each dispenser was cut in pieces and soaked in 25 mL dichloromethane, inside a 50 mL glass centrifuge tube, for 2 h with magnetic agitation. Residual (*E,Z*)-7,9-dodecadien-1-yl acetate contained in the extracts was then measured by GC/FID, using *n*-dodecane as an internal standard. Occasionally, the dispensers were re-extracted to verify the complete extraction of the pheromone.

In the case of the aerosol devices, pheromone release was estimated by the gravimetric method. Six devices were monthly weighted with a scale at Location 2. According to the composition label of the aerosols used, the pheromone is present as a 9.11% w/w. The amount of pheromone released over a period was calculated after applying that percentage to the weight differences.

### 2.5. Efficacy of the Treatments

The efficacy of the MD treatments was checked according two parameters: the captures per trap per day (CTD)—to obtain the male flight curves—and the percentage of infested bunches, obtained with crop damage assessments carried out after each male flight peak.

For capture monitoring purposes, four delta traps baited with a “Grapemone” lure (OpenNatur, Lleida, Spain) and provided with a sticky base were set in the center of each treatment (70 m apart in a square arrangement) in both locations (inside the vine canopy at 1 m above the ground). Although weekly revision of captures provides more robust data on the population dynamics and pest pressure, in this occasion, population was monitored fortnightly in the MD-treated and untreated fields just to evaluate the catch suppression efficacy of each type of MD dispenser compared with the untreated control plots. The Grapemone lures were replaced every 42 days, as a period of efficacy ensured by the manufacturer. The absence of catches in MD plots was considered a necessary condition for MD success.

In addition, crop damage assessment gave the final proof for the treatment efficacy. For this purpose, at least 200 bunches were inspected in 50 vines surrounding each delta trap in each plot in both locations to quantify damaged bunches after each male moth flight peak. In the first assessment a flower cluster was considered damaged when nests, larvae, or pupae were found. In the second and the final assessments a bunch was considered damaged when nests, eggs, larvae, or pupae were found in the visual inspection.

### 2.6. Data Analysis

All the statistical analyses were performed using SPSS v. 16 (Armonk, NY, USA).

The analysis of variance (ANOVA) followed by post-hoc Fisher’s least significant difference (LSD) test at *p* < 0.05 was used to study the differences observed in the absolute airborne pheromone concentrations found throughout the crop cycle in each MD treatment. The analysis was applied to these data without transformation, as they fulfilled the homoscedasticity requirements and the residuals of the ANOVA fitted a normal distribution. In the case of the leave adsorption experiments, data was normalized prior to the ANOVA using √x to fulfill the homoscedasticity and normal distribution requirements.

In the case of population monitoring and crop assessments data a multi-factor ANOVA followed by post-hoc Fisher’s least significant difference (LSD) test at *p* < 0.05 was done. Population monitoring data (male captures per trap and day, CTD) were analyzed after data normalization using the data transformation reported by Stelinski et al. [15], ln(x+1), to fulfill homoscedasticity requirements and normal distribution of ANOVA residuals. These analyses were applied to the whole capture data set and also for each flight peak separately. Data from each crop damage assessment (percentage of damaged bunches) were subjected to the angular transformation $arcsin\left(\sqrt{x}\right)$, prior to the multi-factor ANOVA.

## 3. Results

### 3.1. Airborne Pheromone Quantification

Field volatile collections and the following chemical analysis showed that airborne pheromone concentrations throughout the season varied depending on the MD treatment deployed (Figure 1). The mean concentration obtained with the Isonet dispensers was 4.84 ng/m^3^ in May and the quantity increased up to 42.20 ng/m^3^ in the middle of the crop cycle, significantly differing from the initial concentration (F = 30.739; df = 9, 2; *p* < 0.001). Finally, the last sampling highlighted a decrease in the concentration up to the initial levels of 4.76 ng/m^3^. By contrast, the mean pheromone concentration obtained with Checkmate dispensers was 0.44 ng/m^3^ at the beginning of the season. This concentration also increased significantly (F = 44.273; df = 10, 2; *p* < 0.001) in the middle of the crop cycle (22.34 ng/m^3^). Then, the airborne pheromone concentration with Checkmate dispensers was maintained until the end of the trial in the last sampling period (30.83 ng/m^3^; Figure 1).

The main difference between both releasing systems is that Checkmate Puffer Lb maintained a similar level of airborne pheromone from the middle to the end of the crop cycle, whilst the Isonet L airborne concentration decreased at the end of the crop cycle similar to levels recorded at the beginning of the crop cycle.

### 3.2. Pheromone Leaves Adsorption Capacity

#### 3.2.1. Experiment 1

Immediately after the single aerosol emission, the total amount of pheromone quantified on leaves was 312.4 µg/g, which decreased significantly up to 38.7 µg/g in just 4 h after the application (Figure 2; F = 48.995; df = 3, 15; *p* < 0.001). This amount represented approximately a decrease of almost 88% of the initial content. Then, this content kept on reducing, but at a slower rate, up to 8.3 µg/g in the third sample taken at 17:00 h (8 h after the application). Finally, the last sample taken 24 h after the application showed 5.6 µg/g. Thus, the quantified values suggest that vine leaves can release the previous adsorbed pheromone probably acting as an alternative pheromone releasing sources or regulators for the airborne pheromone concentrations.

#### 3.2.2. Experiment 2

When leaves were treated throughout the evening-night, despite the different mean amounts quantified (383 µg/g, 239.9 µg/g and 177.6 µg/g, 3 h, 7 h and 11 h after the last emission, respectively), these were not significantly different (Figure 3; F = 0.560; df = 2, 13; *p* = 0.587). We observed high variability in the different samples for every treatment, as can be observed in the wide error bars of Figure 3. This could be possibly explained by the different leave position on the plant and the difficulties to establish an “aerosol action area” due to the irregularity and diffusion of the plume and the wind effect. Despite the lack of statistical support, data suggest a similar decreasing trend to that observed in Experiment 1 (Figure 2). Connecting the mean amounts of pheromone with the time passed after the last emission by fitting a linear regression model, *y = a + bx*, (*y* = 390.89 − 28.61*x*; R^2^ = 0.91), the mean pheromone emission from leaves could be established at 28.61 µg/h.

#### 3.2.3. Experiment 3

When leaf samples were taken in the commercial vineyard treated with MD at Location 2, the results indicated that leaves adsorbed similar mean quantities of pheromone (0.07 µg/g) when taken in the center of the plots treated either with aerosols or passive dispensers (F = 1.285; df = 2, 8; *p* = 0.343; Figure 4). In the border of the aerosol-treated plot, this quantity reached a mean of 0.14 µg/g, although this measure had a high variability—the sample collected closest to the aerosol reached 0.26 µg/g, while the furthest contained only 0.06 µg/g.

### 3.3. Dispensers Pheromone Release Profile

#### 3.3.1. Passive Dispensers’ Release Profile

The quantified residual pheromone contents of the Isonet L dispensers fitted the linear regression profile depicted in Figure 5 until the end of the growing season (140 days), when their useful life started to deplete. This means that pheromone load decreased at a constant rate and the slope of the resulting equation gives a mean pheromone release rate of about 1.2 mg/day/dispenser. Thus, the total amount of pheromone emitted was estimated at 526.29 ± 76.30 mg/day/ha from May to September (Table 2). Then, the release rate decreased in the period from early September to mid-October (from 140 to 180 days) up to one third of the previous level (22,685 mg/day/ha).

#### 3.3.2. Aerosol Devices Profile

The resulting release profile of the Checkmate aerosol devices also fitted the linear equation depicted in Figure 6, which means a significantly constant mean release rate of 0.15 g/day/dispenser and a total pheromone emission of 602.49 ± 37.15 mg/day/ha (Table 3). In this case, aerosols did not show a decrease in their emission, in comparison to the passive dispensers, because of the dispenser’ nature.

### 3.4. Population Monitoring

Male catches per trap and day (CTD) recorded throughout the assay (Figure 7) were low at both locations and not significantly different between them (F = 1.684; df = 71, 1; *p* = 0.20), which suggests that the infestation level of both locations was similar. These low catches can be explained due to the use of mating disruption against EGVM in these areas during the last years.

The mean captures recorded in each flight were statistically similar (F = 0.115; df = 71, 2; *p* = 0.89). In addition, both MD treatments significantly reduced moth captures compared to the UTC (F = 12.124; df = 71, 2; *p* < 0.001).

During the first flight, no male captures were recorded in the Isonet L treatments. However, despite a 60% mean inhibition of male catch in the Checkmate treatments, it was not statistically different from the UTC (F = 4.282; df = 23, 2; *p* = 0.09). Captures obtained in both MD treatments were also not significantly different (*p* = 0.29). No male captures were recorded during the second flight at either location with the MD treatments, highlighting the high efficacy performed by both type of dispensers. During the third flight (F = 11.961; df = 23, 2; *p* < 0.001), a statistically significant catch reduction was obtained with both dispensers, Isonet L (*p* < 0.001) and Checkmate (*p* = 0.001) against the UTC treatments, but they did not significantly differ between them (*p* = 0.945).

### 3.5. Crop Damage Prospections.

The effect of location was significant only on the last assessment (F = 4.551; df = 71, 1; *p* = 0.04), when damage levels increased, especially in the UTC. Specifically, the mean percentage of infested bunches was 35% at Location 1, whereas this value decreased to 14.5% at Location 2.

As it is summarized in Table 4, damage assessments carried out in the UTC plots showed an average of about 5% of affected bunches during early and mid-summer (assessments 1 and 2). However, just before the harvest (assessment 3), this value increased up to high levels, exceeding 20%.

The damage was significantly reduced using both MD dispensers compared to the UTC treatment damage (Table 4). Moreover, the percentage damage was not significantly different between the MD treatments in all the assessments done throughout the crop cycle. Thus, both types of dispensers were proven effective, reducing crop damage by 96.97% in the case of Checkmate dispensers and 86.87% in the case of Isonet L dispensers compared to the UTC.

## 4. Discussion

Despite its low concentration and the complexity and diversity of substances present in the environment [42], in this work, we developed a new methodology capable of quantifying airborne pheromone concentrations. Due to the increasing concerns about the impact of semiochemicals into the environment [43], this methodology could be useful to detect and quantify their presence and elucidate potential consequences. In our case, this methodology has allowed the detection of different airborne pheromone concentrations throughout the season between aerosol and passive dispensers’ treated plots.

Although portable EAG can help to understand the variations of airborne pheromone along the day and their dynamics in front of episodes as winds, dispensers’ distribution, etc. [29,30,31,32], it does not provide absolute data. Despite the large sampling periods required (48 h), our results provide actual pheromone concentration measures that can be related with the efficacy of mating disruption treatments (expressed as male flight) providing valuable information for a better understanding of how the pheromone remains in the field and to reduce pheromone wasting. Moreover, it may help to establish a relationship with dispenser characteristics to improve their density and emission and so reducing the cost of implementation of the technique.

The results from the field air collections and quantifications showed differences between the two systems tested for the mating disruption of EGVM. In the case of the passive dispensers, lower mean quantities of airborne pheromone were detected at the beginning of the crop cycle, whereas the concentration reached a maximum in the middle of the cycle during summer, which, among other factors explained below, highlights the dependency on temperature of their release kinetics [44] as can be seen in their emission rates. Similar results were obtained in relative data by Karg et al. [31] in which significant differences were found between EAG measurements done in spring and summer, showing lower EAG signals in the spring studies. Later in the season, the concentration values decreased at the end of the crop cycle, coinciding with the end of the dispensers’ lifespan. The concentration profile was quite different in the aerosol devices treated area, showing a minimum mean concentration at the beginning of the crop cycle, and maintaining higher quantities for the rest of the season. In both cases, the initial low quantity of pheromone detected could be explained by the lack of a developed canopy. When the canopy is scarce, the wind velocity increases and consequently the pheromone may be washed away. In this regard, it has been reported that foliage is able to act reducing the convection air streams keeping the pheromone in the crop environment [31,45]. This effect could explain why the mean pheromone concentrations detected in the aerosol treated areas at the beginning of the season are lower than those achieved with the passive dispensers. The latter system does not stop emitting pheromone during the day and is able to maintain a certain airborne pheromone concentration despite the lack of plant foliage. However, aerosol devices are designed to release the pheromone only at programmed intervals during the evening–night. Thus, airborne pheromone is not continuously supplied, making this system more sensitive to pheromone washing. This phenomenon agrees with the lower male flight inhibition (first flight) achieved in the aerosol devices treated areas, as shown by our monitoring data and, also, by other technical reports [46,47]. The high number of passive dispensers installed per hectare makes the competitive attraction ‘females-dispensers’ possible, preventing males from female finding. In contrast, the low density of pheromone sources in the case of aerosols and the lack of canopy might lead the pheromone to the soil, where the degradation could be triggered as described by Shaver [48].

The highest mean airborne pheromone concentration was obtained in the second volatile sampling (July) with both emission systems coinciding with the highest mean temperatures. When these samplings were performed, all vines had a well-developed canopy, supporting the hypothesis of the role of the canopy in regulating field air streams. Moreover, several studies describe leaves as pheromone sources (by adsorption and release) [49,50,51,52], based on the reported adsorption of the pheromone components on their waxy surfaces [50]. However, their interaction with the airborne pheromone only was studied in a few cases [23,53,54]. This “adsorption-releasing” mechanism and their interaction with the airborne pheromone is supported by the results of our experiments. When the aerosol was shooting over the vines during the whole night (Experiment 2), leaves sampled 3 h after the last emission still had 383.1 µg/g in their surface. Then, a decreasing trend in the quantity of pheromone was observed throughout the day, suggesting that pheromone is released from the leaf surface at an estimated rate of about 28.6 µg/h per gram of leaves. This value is comparable with those obtained in similar studies done by McGhee [17], in which 15 apple leaves receiving five emissions from a CM MIST emitted codlemone at 4 µg/h in apple orchards. In our Experiment 1, we also observed that this release rate could be higher immediately before the aerosol emission, as the quantified amounts of pheromone decrease from 312 to 38 µg/g in 4 h (approximately 68 µg/h). Despite the lack of detection of degradation products of the pheromone in the leaf experiments, leaves are not probably free of these degradation processes as they are also exposed to climate conditions, as it occurs on the soil. These adsorption-releasing dynamics are not exclusive of the aerosol devices system, as the leave samples collected in the area treated with passive dispensers (Experiment 3) showed similar quantities of pheromone (70 ng/g) than those collected in the aerosols area. Considering that the quantity of pheromone emitted by *L. botrana* females has been estimated near 0.3 ± 0.1 ng/h, as a maximum in the first hour of the scotophase [55], and the amount of pheromone adsorbed on leaves and their proved dynamics, our data supports that vine leaves could imply countless pheromone point sources capable of competing with females.

During the second sampling (end of July), the airborne pheromone concentration was significantly higher in the area treated with passive dispensers than in the aerosols plot. Taking into account that the canopy is acting in both MD treatments and the release profiles of both systems are similar; this difference can be attributed to the distribution of the pheromone sources. Given that the volatile collections are performed in the center of each plot, it is possible that the passive dispensers ensure a better airborne pheromone distribution inside the plot. Nevertheless, the airborne pheromone concentrations obtained in both MD treatments provided effective disruption levels (total male capture inhibition compared to control plots).

When the end of the crop cycle was near (September), airborne pheromone concentration with the passive dispensers dropped to initial levels, which agrees with their reduced mean emission rate (526.29 mg/day/ha vs. 226 mg/day/ha) coinciding with the end of their lifespan. At this moment, the quantity of pheromones emitted by the dispensers is too low and the adsorbing and post-releasing mechanism of the leaves is not able to compensate airborne concentrations because the adsorption-release dynamics of the leaves might be too quick [31,32]. On the contrary, aerosols were able to maintain high airborne pheromone concentrations because their emission is programmed.

The distribution of the quantified airborne concentrations throughout the crop cycle suggests that the plant canopy is not a requirement to achieve disruption when using passive dispensers. However, it helps keeping the pheromone within the field area by reducing pheromone washing, distributing the pheromone more uniformly and increasing the number of pheromone sources along the crop, and thus improving the competitive mechanisms of disruption. On the contrary, the automatic aerosol system is somewhat canopy-dependent. As obtained in the Experiment 3, leaves randomly sampled in the center of the plot treated with aerosols significantly contained the same amount of pheromone than those from the center of the passive dispensers’ plot. This might explain how mating disruption works when installing aerosols, as leaves may be acting as additional point sources within the vineyards. Moreover, the variability in the data quantified in samples from the border of the aerosol plots, highlighted by the wider error bars, might respond to a gradient of pheromone content according to the distance to an aerosol. The nearer the leave samples from an aerosol device, the higher the pheromone quantified from those leaves. This progression in the amount of pheromone adsorbed on the leaves according to distance would be the result of the plumes generated by the aerosols, which can reach more than 460 m long and 90–150 m wide as Welter et al. [7] outlined. In this way, the nearest vines of the aerosol devices would act as high releasing dispensers during the periods between the emissions. These results support the reported theory of induced allopatry in which males are attracted to the pheromone sources following an increasing concentration of pheromone when using aerosol devices [9,16,17]. At the same time, leaves loaded gradually along the distance would act as temporary plumes between aerosol emissions.

Taking into account male flight inhibition values in our trials and the different pheromone airborne concentrations found, we might estimate that the lowest airborne pheromone concentration in which male flight was effectively disrupted could match the mean concentration found in vineyards treated with passive dispensers at the beginning of the growing season, approximately 5 ng/m^3^. The inhibition values would confirm that mating disruption might be not properly performing as such in our assays with pheromone concentration of approximately 0.5 ng/m^3^. Further trials would be necessary to reduce this range of values and determine an optimal pheromone concentration in air. Considering competition as the main mechanism acting in mating disruption treated fields and leaves acting as countless pheromone sources, overcoming this concentration could result in pheromone wasting. Nevertheless, different pest populations and different conditions such as climate, vineyard shape, pheromone distribution along the field, etc., may affect the results and this could be considered in further trials.

## 5. Conclusions

In the present work, a new methodology based on high volume air samplings and current sensitive chromatographic and spectroscopic methods—GC/MS/MS—was developed to detect and quantify airborne pheromone concentrations. This new methodology was employed to gain insight into the mode of operation of two different releasing systems—passive dispensers and aerosol devices—employed to apply MD against EGVM. The results found could be used to optimize and reduce pheromone waste in MD treated fields.

Although both pheromone dispensing systems released the same average quantity of pheromone and displayed similar control of fruit damage, spatial and seasonal differences in pheromone distribution were demonstrated. Passive dispensers kept lower airborne pheromone concentrations at the beginning and the end of the crop cycle with a maximum in mid-season. The treatment significantly reduced male catches compared to non-treated vineyards during the whole crop cycle. By contrast, aerosol devices were not capable of fully inhibiting male catches at the beginning but kept effective high amounts of airborne pheromone at the end. The canopy role as pheromone concentration regulators—adsorbing and releasing previously emitted pheromone—was demonstrated and it is suggested as an important factor affecting the efficacy of the aerosol devices. In addition, this capacity would explain how competitive mechanisms act when using both types of dispensers. Airborne pheromone data collected demonstrates that in our experimental conditions, airborne pheromone values just over 5 ng/m^3^ effectively inhibited male captures, reflecting a good performance. Although this must be proven in similar studies made in locations with different climate conditions and pest-density populations, our results highlighted that there is still room for improvement to optimize pheromone use. Similar efficacies might be obtained with lower emission rates and lower initial pheromone content in the case of passive dispensers, or adjusting the releasing time of automatic aerosol devices, in areas with low pest pressure or with mating disruption backgrounds. Moreover, it could be interesting to test if a delayed deployment of aerosol devices after the first flight could obtain the same efficacy.

## Figures and Tables

**Figure 1 insects-11-00289-f001:**
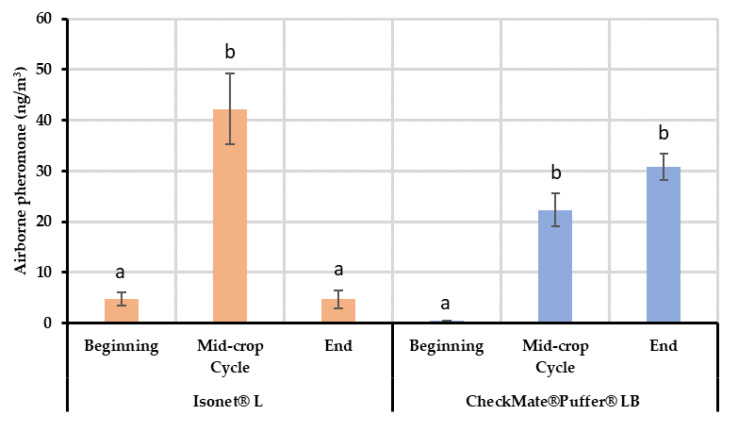
Mean (± SE) absolute airborne pheromone concentrations (ng/m^3^) in vineyards treated with passive dispensers (Isonet^®^ L) and aerosol devices (CheckMate^®^ Puffer^®^ LB). Air samples were collected at three moments during the crop cycle: (1) the beginning of the crop cycle (second week of May to middle June), (2) in the middle of the crop cycle (last week of July–first week of August) and (3) at the end of the crop cycle (second to third week of September). Data of each type of dispenser were analyzed separately by applying an ANOVA followed by the Fisher least significant difference (LSD) test (*p* < 0.05). Bars labeled with different letters for each type of dispenser are significantly different.

**Figure 2 insects-11-00289-f002:**
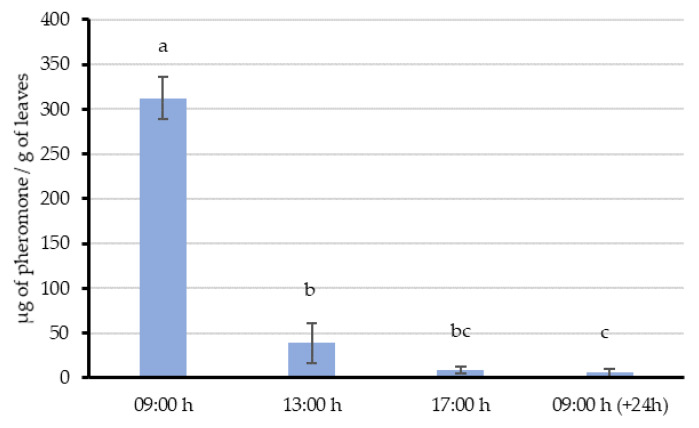
Mean (± SE) quantity of pheromone (µg pheromone/g leaves) adsorbed on leaf samples of Experiment 1, after one aerosol emission. Bars labeled with different letters are significantly different, ANOVA followed by the Fisher LSD test (F = 48.996; df = 3, 15; *p* < 0.001).

**Figure 3 insects-11-00289-f003:**
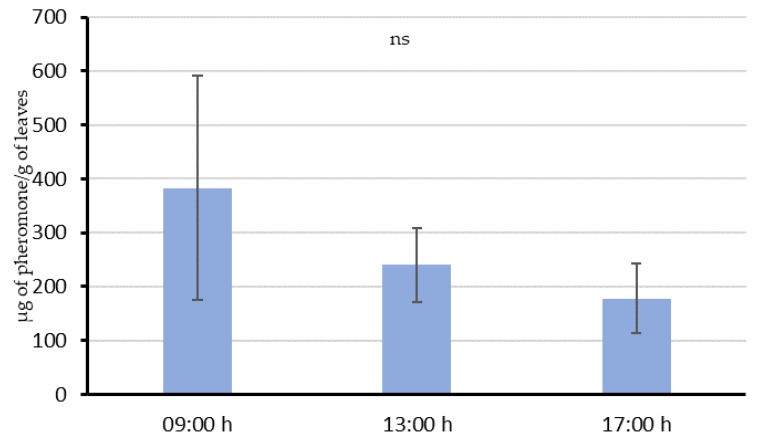
Mean (± SE) quantity of pheromone (µg pheromone/g leaves) adsorbed on leaf samples of Experiment 2, after overnight aerosol shooting (from 18:00 to 6:00). Bars labeled with different letters are significantly different, ANOVA followed by the Fisher LSD test (F = 0.560; df = 2, 13; *p* > 0.05).

**Figure 4 insects-11-00289-f004:**
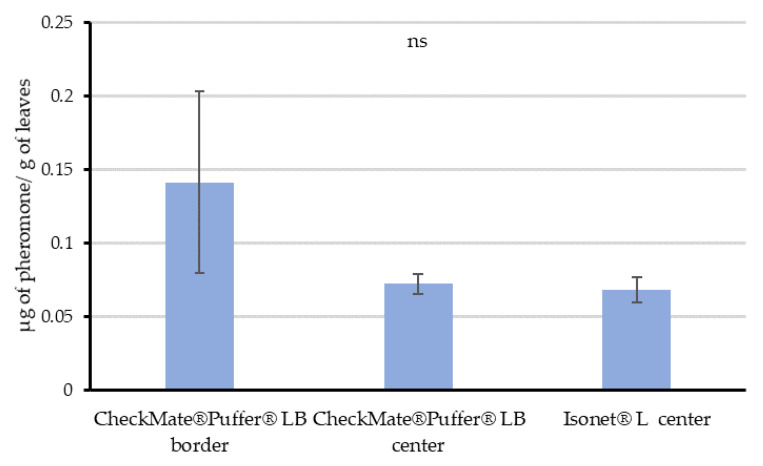
Mean (± SE) quantity of pheromone (µg pheromone/g leaves) adsorbed on leave samples of Experiment 3, collected in actual field conditions at Location 2: along the row where Checkmate devices were deployed (aerosol devices border), 70 m away from the aerosols row (aerosol devices center) and at the center of the Isonet passive dispensers treated field (passive dispensers center). Bars labeled with different letters are significantly different, ANOVA followed by the Fisher LSD test (F = 1.287; df = 2, 8; *p* > 0.05).

**Figure 5 insects-11-00289-f005:**
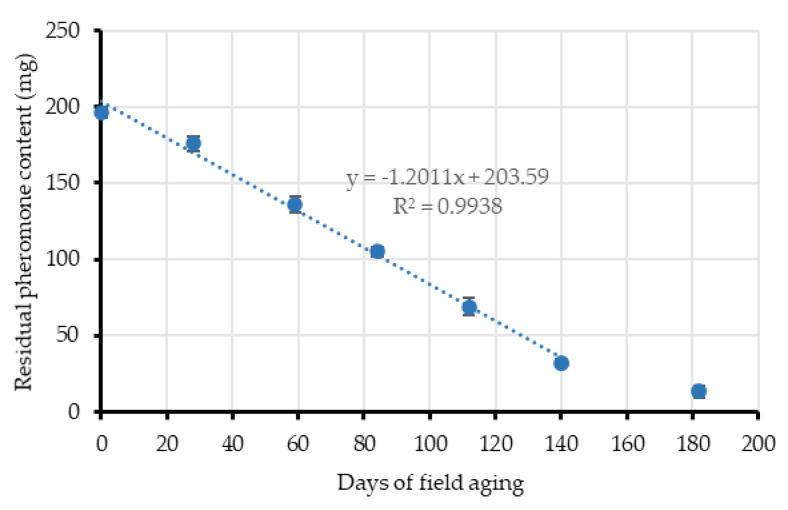
Pheromone release profile of Isonet L passive dispensers, as residual mean pheromone load (mg) contained in the dispensers vs. the time of field exposure (days). Release kinetics fitted the linear model given by the equation.

**Figure 6 insects-11-00289-f006:**
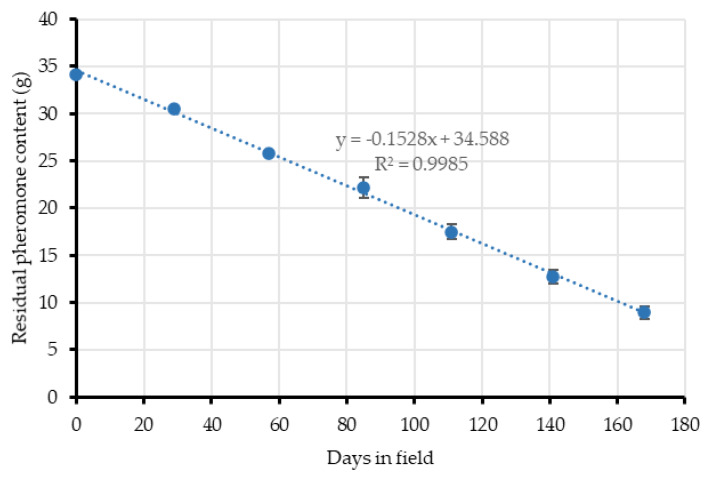
Pheromone release profile of Checkmate aerosol devices, as residual mean pheromone load (mg) contained in the dispensers vs. the time of field exposure (days). Release kinetics fitted the linear model given by the equation.

**Figure 7 insects-11-00289-f007:**
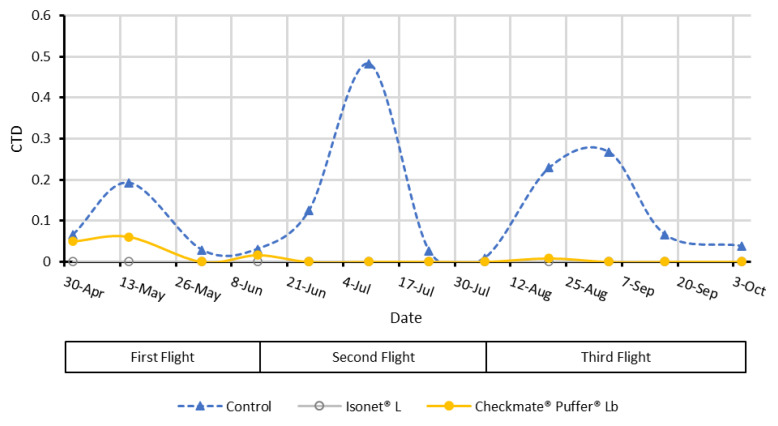
Mean of male captures per trap per day (CTD) throughout the season at Location 1 and 2 in the plots with different treatments.

**Table 1 insects-11-00289-t001:** Optimum values of the MS/MS parameters for each target compound.

	Transition	Precursor Ion (*m/z*)	Product Ion (*m/z*)	Collision Energy (eV)
TFN	1	375.0	263.0	10
	2	393.0	373.0	5
EGVM pheromone (main component)	1	79.1	77.0	10
	2	93.1	77.0	10
	3	164.2	79.0	15
	4	164.2	94.1	5

**Table 2 insects-11-00289-t002:** Isonet L passive dispensers aging. Residual means of pheromone content (mg) throughout the time in field (days). Point values of Figure 5 and the total mean of pheromone used (mg/day/ha).

Days in Field (d)	Pheromone Content (mg)	Dispensers/ha	Pheromone Emission (mg/day/ha)
**0**	196.02 ± 3.39	500	-
**28**	176.04 ± 4.74		356.71
**59**	135.82 ± 5.58		648.67
**84**	105.08 ± 2.92		614.85
**112**	68.81 ± 5.69		647.80
**140**	31.68 ± 3.03		662.89
**182**	12.63 ± 4.03		226.85
		**Mean**	**526.29 ± 76.30**

**Table 3 insects-11-00289-t003:** Checkmate aerosol devices pheromone content. Residual means of pheromone content (mg) throughout the time in field (days). Point values of Figure 6 and the total mean of pheromone used (mg/day/ha).

Days in Field (d)	Weigh (g)	Pheromone Content (9.11% w/w; g)	Devices/ha	Pheromone Emission (mg/day/ha)
**0**	375.25 ± 0.75	34.19 ± 0.07	4	-
**29**	334.75 ± 0.75	30.50 ± 0.07		508.90
**57**	282.75 ± 1.70	25.76 ± 0.16		676.74
**85**	243.75 ± 12.09	22.21 ± 1.10		507.56
**111**	191.83 ± 8.66	17.48 ± 0.79		727.63
**141**	139.83 ± 8.11	12.74 ± 0.74		631.63
**168**	98.16 ± 7.53	8.94 ± 0.69		562.48
			**Mean**	**602.49 ± 37.15**

**Table 4 insects-11-00289-t004:** Mean (± SE) percentage of damaged bunches obtained in each treatment in the three damage assessments carried out at Location 1 and 2.

Assessment ^1^	Damage (%) ^2^			Stats ^3^			
	Control (C)	CheckMate Dispensers (CM)	Isonet Dispensers (I)	Comparisons	F	df	*p*
1	5.00 ± 1.19 (*n* = 400)	1.00 ± 0.53 (*n* = 400)	0.75 ± 0.53 (*n* = 400)	C-CM *	24.009	71, 2	<0.001
				C-I *			<0.001
				CM-I			0.331
2	4.39 ± 0.66 (*n* = 430)	0.91 ± 0.49 (*n* = 420)	0 (*n* = 420)	C-CM *	14.753	71, 2	<0.001
				C-I *			<0.001
				CM-I			0.539
3	24.75 ± 6.64 (*n* = 400)	0.75 ± 0.37 (*n* = 400)	3.25 ± 1.60 (*n* = 400)	C-CM *	15.058	71, 2	<0.001
				C-I *			<0.001
				CM-I			0.781

^1^ Damage assessments were performed after the first, second and third male flights. ^2^ Percentage of damaged bunches. ^3^ Results of the statistical analysis (ANOVA test, at *p* < 0.05), statistical differences are marked with asterisks.

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
