# Peer review of "Airborne Pheromone Quantification in Treated Vineyards with Different Mating Disruption Dispensers against Lobesia botrana"

_insects, 2020, doi:10.3390/insects11050289_

Round 1

Reviewer 1 Report

The paper “Airborne Pheromone Quantification in Treated 2 Vineyards with Different Mating Disruption 3 Dispensers against Lobesia botrana” is an excellent study into examination of pheromone concentrations in vineyards treated with two different emitting systems and the effect on EGVM behavior and resulting infestation. The authors conducted elaborate tests to quantify airborne concentrations of pheromone as well as absorbance to foliage. Despite the importance of the findings, there are several major problems with the manuscript as written in it’s current form.

Statistical analyses and the rules that define the tests are critical in any research. The authors need to address the major question of replication of treatments. It appears (at times) that there are two of each treatment, one of each at 2 separate locations called “Fields”. At other times it seems as if only 1 treatment is at a given location (in experiments on airborn concentrations of pheromone and leaf absorbance). Proper statistical analyses should combine the data from each replication (location) for comparison of treatments.  The authors have reported each site independently comparing all three treatments.  

The paper should not be published until a clear explanation of experimental design is presented for the contained experiments and the data analyses are properly addressed.

Specific comments follow:

L22      buffers? Alternative pheromone sources

L 54     pest control technique… (rearrange wording)

L59      metal can… (both aluminum and steel are used by several companies)

L60      pheromone, diluent, and propellant…

L62      via emissions (replace shots)

L71      (reword). Currently, growers can disrupt EGVM using either passive pheromone dispensers or aerosol releasing devices.

L78      replace spread (increase)

L110    replace nowadays (currently)

METHODS

The experimental design in the Methods is not reflected in the RESULTS Section. The METHODS are currently very confusing with no experimental design indicated.

Currently, the Methods read accordingly:

Field 1 - Venta Del Moro (Treatments - Isonet EGVM, UTC)

Field 2 - Moixent (Treatments - Checkmate Puffer LB, UTC)

Is this correct? If so, then there is no replication of treatments and any effects comparing MD treatments to each other is invalid.

On the other hand, if each location had all three treatments statistical analyses comparing MD treatments to one another are valid:

Field 1 - Venta Del Moro (Isonet EGVM, Checkmate Puffer LB, UTC)

Field 2 - Moixent (Isonet EGVM, Checkmate Puffer LB, UTC)

Replication is limited at 2, but acceptable for large field trials in many cases, especially when measurements are taken over time.

In this case the methods should clearly state something like; “The experiment was a complete block design with 2 locations (block = Venta Del Moro and Moixent) and 3 treatments within each block (Isonet EGVM, Checkmate Puffer LB, and UTC).

Currently, the manuscript as written, does not have replication (of treatments) other than the UTC.

Please refer to the “Fields” as LOCATIONS (location/Site 1 and Location/Site 2) and the treatments as (Isonet, Checkmate, UTC).  Eliminate the use of “plot” and “field” as they carry no useful meaning when jumping back and forth between sites/Locations.

L139    replace shots (emissions)

L140    each quarter hour….

Section 2.2.3 Field air sampling

L228    replace leave with Leaf

L230    2m between “shots”. This is the same distance the aerosol emissions were measured from. It seems pretty close between replications. i.e. there was no buffer between reps?

L230    replace shots with emissions

Indicate where Checkmate puffer elevations in reference to the vineyard floor and canopy. Were they level with the vineyard foliage? Above, below?

L232    replace shots with “an emission”

L233    replace shots with “an emission”

L234    replace done with conducted

L286 delete performance.

L336    acting as a buffer? Or alternative pheromone release sources. Buffers are typically considered neutral areas that separate 2 treated areas.

3.3 Population monitoring

Methods and Results Sections are incongruous as stated previously.

RESULTS - figures 7 & 8 indicate Isonet EGVM, Checkmate Puffer LB, and UTC at each “Field” (Location?) but this is not how the current METHODS are written.

L400-402. It appears that the authors are comparing male captures in MD treated plots (Isonet and Puffer). Concluding infestation levels (numbers of EGVM) were the same at each site/location/block cannot be established when the comparison is based upon an influencing (MD) treatments.  Comparing the overall infestation or population level of EGVM between locations/sites/plots can only be conducted in the UTC treatments where they are NOT influenced by pheromones.

L400 “Male catches per trap and day recorded throughout the assay were low in the UTC treatments at both locations and not significantly different between them……

L404 “The most abundant peak of EGVM occurred during the third flight at location 1 and during second flight at location 2 (figures 7 & 8)”.

L406    No male moths were captured in the Isonet EGVM treatment at location 1 and no moths were recorded in either Isonet or Checkmate during second flight at either locaiton.

L419    In Methods field 1 is listed as Isonet hand-applied dispensers and field 2 is aerosol dispensers. In this section they are reversed. Is “field 1” and “field 2” the treatment or the location?

L419    at location 2 Isonet EGVM completely inhibited male flight compared to the UTC

L436    Field 1 & Field 2 (replace with Location 1 and Location 2) if this is correct.

L437    damage level at location 1 was statistically …..

L451    At location 2, Isonet EGVM reached….

L476    …”have made it necessary to gain more insight into the ….”

L499 – 517      discussion on passive Isonet EGVM dispensers. Might the decrease in the airborn concentration be more due to the fact that relative temperatures have decreased?  Low concentrations were found at both the beginning and end of the season when temperatures are colder than the summer.  Generally these Isonet type ropes emit pheromone on a zero order, meaning they release at a constant rate over time with the exception of influence by temperature.  Other passive releasing devices release on a first order kinetics, where they blow off large amounts of pheromone and the rate of loss decreases as the season progresses.

See (Hebert, V. R., Tomaszewska, E., Brunner, J. F., Jones, V. P., & Doerr, M. (2007). Evaluating the pheromone release rate characteristics of commercial mating disruption devices.)

L517    canopy (might) lead the pheromone to the soil….”L545       actign on both MD treatments

L549    obtained in both MD treatments

L554    again clarify how “buffer” is used.

L599    “Although both pheromone dispensing systems released the same….”

Figures and Tables

Be consistent with decimals or commas. Commas are typically used for separating units by thousandths. i.e Table 3, days in field 0,  23.19 +- 0,07. Review all figures and tables.

In the case of air sampling at three different vegetative state; 4 samples each beginning, 4 samples aerosols and 3 samples Isonet mid-crop, and 3 samples each end. Were these samples taken at each location? Again, I refer to replication. Are all three treatments at each location?  IF yes, then the collection of samples should be thus:

Scenario A – replication of treatments

Treatment

Sample timing

Location 1

Location 2

Aerosol

beginning

4

4

Isonet

beginning

4

4

UTC

beginning

0

0

Aerosol

Mid

4

4

Isonet

Mid

3

3

UTC

Mid

0

0

Aerosol

End

4

4

Isonet

End

4

4

UTC

End

0

0

Scenario B – No replication of treatments

Treatment

Sample timing

Location 1

Location 2

Aerosol

beginning

0

4

Isonet

beginning

4

0

UTC

beginning

0

0

Aerosol

Mid

0

4

Isonet

Mid

3

0

UTC

Mid

0

0

Aerosol

End

0

4

Isonet

End

4

0

UTC

End

0

0

Scenario C – replication of treatments with half of reported samples coming from each location.

Treatment

Sample timing

Location 1

Location 2

Aerosol

beginning

2

2

Isonet

beginning

2

2

UTC

beginning

0

0

Aerosol

Mid

2

2

Isonet

Mid

2?

1?

UTC

Mid

0

0

Aerosol

End

2

2

Isonet

End

2

2

UTC

End

0

0

Proper statistical analyses would compare the average of results from location 1 & 2 for each treatment vs the other treatments for each time period (beginning, mid, end), (Scenario A).  The question remains, did all samples for a given treatment come from only 1 treatment in 1 location (Scenario B)? This is replication of sampling and not replication of treatment. I again differ that papers are published using this methodology, but it breaks the rules of statistical analyses. Did 2 samples each come from each treatment at each location to equal 4 total treatments as reported- Scenario C?  Furthermore, it would have greatly benefited the manuscript tot include samples from an untreated check. As researchers we often assume that what we do not expect is not worthy of measurement, especially in this scenarios where labor and air sampling is very challenging, but such a sample would absolutely confirm the absence of pheromone influence as expected. Conversely, if the experiment for any reason detected pheromone in the UTC these baseline values could be deducted from the MD treatments to measure the actual influence from those treatments.

When using only a single treatment replication of measures over time is valid, but only when comparing these time measurement within that single treatment. Comparison between treatments is not acceptable in this scenario. However, the Authors could discuss the relative differences at the end of the manuscript and the implications of those results if the data when replicated remained the same.

Release profiles of emission systems

Figure 1. Comparison within a treatment is valid. Comparisons across treatments is not valid without location of the treatment.

Figure 4. Comparison within a treatment is valid. Comparisons across treatments is not valid without location of the treatment.

Figure 7, 8, Table 4, 5 Comparison within a treatment is valid. Comparisons across treatments is not valid without location of the treatment.

At times it appears that there is replication of all 3 treatments at two locations; other times it does not.  The authors need to clearly address the experimental design for each experiment if the experiments differed in any way. i.e. air sampling conducted in only 1 treatment (no treatment replication), but samples replicated over time. Valid to compare time results in 1 treatment but not across treatments.

In the later experiments where Field 1 and Field 2 are reported separately (Figure 7, 8, Table 4, 5), it appears that all 3 treatments are at each location.  In this case the data should be pooled between locations (Field 1 &2) and analyzed across treatments (Isonet, Checkmate, UTC) with location as a blocking effect in the analyses.

Author Response

Please, find attached in a Word document the reply to your report.

Reviewer 2 Report

The work concerns the interesting aspect of the detection of the concentration of synthetic sexual pheromone (SSP) in the field following the application of the mating disruption technique (MDT) against the European grapevine moth, Lobesia botrana (EGVM).

This primary objective was complemented by that of evaluating the effects of estimated SSP concentration levels, resulting from the application of two different MDTs (Shin-Etsu, Isonet L, 500 dispensers per ha; Suterra, CheckMate Puffer LB, three aerosol devices per ha) on the inhibition of catches of EGVM males by traps triggered with the same synthetic pheromone (Grapemone-baited delta traps) and on the inhibition of mating, indirectly assessed through the effects of the MDT on the infestation of the bunches.

The work appears to consist of two sections. The first and most important section, clearly indicated by the title, is essentially chemistry; the second section, which characterizes the context and the final purpose, is less relevant in the development and articulation of the manuscript and essentially concerns entomology.

While the part of “chemistry”, although requiring improvements in terms of lightening and synthesis, is for me acceptable with a minor revision, the part more closely related to “entomology” has significant shortcomings in terms of contextualization, methodological and experimental approach, data collection in the field, interpretation of the results, their comment from both a scientific and technical point of view.

More precisely, I indicate below and in order of presence in the text the main deficiencies found in the manuscript:

Title

Lines 4 (and: 20, 25, 45, 49, 74, 77, 125, 128, 181, 395, 409, 437, 439, 474, 608) - The authors correctly use the term "passive dispensers" for the "passive reservoir pheromone releasers” and the terms "aerosol devices" for the "automatic aerosol releasers”, but they use the term "dispensers" for both the types. Since the terms "devices" or "releasers" or "diffusers" have a broader meaning of “dispensers”, I think it is more appropriate to use one of these to indicate both formulations.

Abstract:

Line 13 - It is recognized and accepted that EGVM was first described in 1775 by Denis & Schiffermülle as Tortrix botrana, therefore the international taxonomic nomenclature code requires that the authors are indicated in brackets.

Line 20 – Replace “in controlling” with: “in limiting”

Keywords:

Line 27 – Delete: “Lepidoptera”, as it is redundant

Line 28 – Add: "pest control" and possibly also "field trials"

  1. Introduction

Lines 37-39 –The sentence: “This is important for species that are currently controlled by this method and even to introduce its use for species in which the cost used to be too high to be assumed by the crop” is twisted. Reformulate it better!

Line 43 – Replace: “the dispensers’ emission rate” with: "the pheromone emission rate”; and: “their number required” with: “the number of dispenser required”

Line 44 - Replace: "All these parameters depend on" with: "All these parameters are related to"

Line 56 - Since the recent passive reservoir dispensers "Shin-Etsu, Isonet LTT" are usable at an average density of 250 units per ha, I think it is appropriate to replace the expression with: "of more than 250 units/ha" and add a relevant reference.

Line 68 - Replace with: "Denis & Schiffermüller” with: “(Denis & Schiffermüller)” and add: "(Lepidoptera: Tortricidae)"

Lines 68-69 – For the statement: “Lobesia botrana ... has been traditionally controlled using mating disruption” is the reference 16 (McGhee et al. - Optimizing aerosol dispensers for mating disruption of codling moth, Cydia pomonella, L.), relevant???

Line 70 – Add after the expression “passive dispensers at high densities”: “(500 or more recently 250 units on average per hectare)”

Line 81 - None of the references [5, 22-24] concerns Lobesia botrana: consider whether to add at least one of the many, referring to EGVM

  1. Materials and Methods

Lines 132-141- Here there is a very lacking and misleading description of the investigations (trials, tests) carried out in 2018 in the two experimental fields.

In fact, from it seems that the MDT by means of passive dispensers has been applied in a "2-ha plot of vineyard" (Field 1, Valencia, Venta del Moro, UTMs 39.482608 - 1.318822, Bobal variety, about 2667 vines per ha, Shin-Etsu, IsonetL, 500 dispensers per ha, installed on 24 April 2018) and the other MDT by means of aerosol devices has been applied in another "2-ha plot of vineyard” (Valencia, Moixent, UTMs 38.808779 - 0.815917, Monastrell variety, about 2667 vines per ha, (Suterra CheckMate Puffer LB, three aerosol devices per ha, 23 April 2018, via shots each quarter from 18:00 h to 06:00 h).

In reality, the “Results” suggest that the two MDTs were compared with each other both in Field 1 and in Field 2, as well as, in each field, compared with an “untreated control”, consisting of a vineyard plot about 500 m away from the treated plots.

But: 1) nothing is said of how the two MDTs have been set up in each of the two vineyard plots of Field 1 and of Field 2, and if the MDT for the control of EGVM has been integrated by specific or generic insecticide applications; 2) nothing is said of the extent and agronomic, cultivation and varietal characteristics of the two control plots; 3) for these two control plots, nothing is said if and how the possibly control (specific or generic) of EGVM was made; 4) for these same control plots, nothing is said of the type and number of pheromone traps used for the monitoring of the EGVM males, nor of the number of clusters examined for the detection of EGVM infestation.

Furthermore, while the first sentence of the "Materials and Methods" (132) states that “Studies were conducted between March and October 2018, the sixth (136) and the eighth sentence (139) of the same paragraph specify that the two MDTs were applied respectively April 24 in Venta del Moro and April 23 in Moixent. Apart from the obvious contradiction, the application of the MDT appears to be very late in relation to the first flight of EGVM in Valencia province.

Finally, the authors do not justify the choice to consider Isonet L distributors, applied at 500 units/ha instead of the more advanced Isonet LTT dispensers, used at 250 units/ha.

Lines 281-282 – The sentence: “Population was monitored to check the efficacy of each type of dispenser and relate efficacy of mating disruption with airborne pheromone concentration measured in the same fields” is twisted. Population monitoring may have the purpose of evaluating the effectiveness of the treatment but cannot directly relate this efficacy to the concentration of pheromone in the air. Break the period and improve the logical sense. Furthermore, in experimental and statistical terms, the effectiveness of any treatment can only be assessed by comparison with other treatments, or untreated controls! And this concept is too implied and taken for granted!

Line 283 - Specify that the same type and number of pheromone sticky traps were used in each treated and untreated vineyard plot considered in Venta del Moro (4x3) and in Moixent (4x3)

Line 285 - Monitoring of EGVM males worthy of rigorous field trials in this context cannot have such a low frequency of catch control and dispenser replacement in pheromone traps (14 and 42 days respectively).

Lines 285-286 - The statement: “The absence of catches in mating disruption plots was considered as mating disruption success performance.” is incorrect! The failure to catch males of the target species (moth) with pheromone traps has always been logically considered the necessary but not sufficient condition to achieve success from the application of the MDT

Lines 288-289 - Specify exactly how many clusters were examined per each vineyard plot and per each EGVM generation. Was the total amount of the bunches examined equal to 3,600 (50x4x6x3)?

Line 289 – With reference to: “A bunch was considered damaged when eggs, nests or larvae where found” I'm curious to know how you would have been able to detect EGVM eggs at the end of the first flight on the inflorescences in fruit set and why the pupae were not considered as possible stages present in inflorescences and clusters. Moreover the percentage of infested (affected) bunches is a much less precise variable than the number of larvae or nests per bunch (or per 100 bunches), which obviously flattens the differences between the results of one treatment and those of another. Why did you choose to use such an approximate dependent variable?

  1. Results

Line 372 – Replace: “Passive dispensers’ release profile” with: “Passive dispensers release profile”

Line 387 – Replace: “Aerosol devices’ profile” with: “Aerosol devices profile”

Lines 400-402 - The calculation of the number of catches of males per trap per day based on the control of the traps carried out every 14 days is very artificial and not very correct! Furthermore, as already mentioned, the variability of the infestation is not explained statistically by the variability of the catches of the males with pheromone traps, in particular in the vineyards subjected to MDT.

Line 403 - How is it possible to speak of “population dynamics” with such very low catch data even in the control plot?

Lines 420-422 - The sentence: " The absence of captures along the peaks was reflected in statistical differences against the control plot male captures in the first (H = 6.650, P < 0.05), second (H = 12.754, P < 0.05), and third flight peak (H = 11.713, P < 0.05).” is as emphatic as it is trivial.

Lines 433-436 – With reference to the infestation data summarized in Table 4 and Table 5 and commented in these lines, why did you analyse the infestation data separately by trial location? Was it not preferable to analyse the infestation in relation to the control, not only as a result of the treatment, but also of the variety and the environment in general?

Lines 451-452 – With reference to the sentence: “In Field 2, passive dispensers’ treatment reached 100% efficacy in all the assessments throughout the crop cycle.” it must be said that all this emphasis is not justified because the authors should know from the literature that in these contexts 100% effectiveness is practically impossible and that it often depends on an insufficient size of the sample of bunches considered and on the extent of the control infestation.

  1. Discussion

Lines 474-479 - Redundant as already mentioned in the introduction and therefore to be deleted.

Lines 482-483 - Redundant as already mentioned in the introduction and therefore to be deleted.

Lines 492-502 - Avoid repeating data and considerations already provided in the results, which need only be discussed.

Lines 538-539 - Use Italic caracter for L. botrana.

Lines 551-557 - Avoid repeating data and considerations already provided in the results, which need only be discussed.

Lines 578 -581 - Avoid repeating data and considerations already provided in the results, which need only be discussed.

Lines 590-591 – The sentence: “Nevertheless, this kind of studies should be carried out in areas with different pest populations and different climate conditions to draw more powerful conclusions.” sounds like a heavy self-criticism in contradiction with the goal of the research and the Materials and Methods used. Change it appropriately!

Lines 548-550 - Why is the concentration of the pheromone never related to the surface and shape of the parcel/vineyard treated, both parameters that have always been considered of great importance?

  1. Conclusions

Line 602 – “Replace: "in mid-season, the treatment" with: "in mid-season. This treatment"

Line 613 - The population density in the "untreated" control vineyards and other elements that emerged in the field trials lead us to think that the adult population of EGVM present in the plots subjected to MDT was rather modest. This must relativise the consistency of the results obtained and avoid an overly optimistic view of the possibility of reducing, tout court, the amount of pheromone in the field, which, on the other hand, may be susceptible to a better spatial and temporal distribution

Round 2

Reviewer 1 Report

The authors have made great improvements to the original manuscript  especially in addressing the methods and statistical analyses.  There are still some minor grammatical edits necessary prior to publication. All in all the science explored and results presented are very interesting. 

Consider replacing “releasers” throughout the paper with “dispensers”. This is the most common term used for the past 30+ years in MD research. If there was not a physical device, such as microencapsulated flowable pheromones, “releasers” would be sufficient.

Please eliminate "plots" where possible. This suggests that you are comparing physical places rather than the treatments applied to the crops. A common error in many publications. You are measuring the effect of treatment not the plot. Use plot when discussing what happened in that physical space. i.e. “all treatment plots received the same pest management regimes throughout the season.”

L310 – “Population was monitored in the MD-treated and untreated plots to evaluate the catch suppression efficacy of each type of MD releaser compared to the untreated control.

L310- replace “check” with “evaluate” or “measure”. Check (verb) gets too confusing when also using the term check as a treatment (noun).

L319- …treatment plot at both locations to quantify damaged bunches after each male moth flight peak.

L344- delete differently, this is redundant with varied.

L355 – The main difference between the two pheromone releasing technologies indicated that Checkmate Puffer LB maintained a similar level of airborne pheromone from the middle to the end of the crop cycle, whilst the Isonet L airborne concentration decreased at the end of the crop cycle similar to levels recorded at the beginning of the crop cycle.

L398 replace "highlighted" with indicated

L447- “In addition, both MD treatments significantly reduced moth captures compared to the UTC.”

L452- However, despite a 60% mean inhibition of male catch in the Checkmate treatments, it was not statistically different from the UTC.

L455- replace "both" with "either" location

L456- Replace “In the case of….” with “During”

L460- The effect of location was significant only on the last assessment, when damage levels increased, especially in the UTC.

L462- “Specifically, the mean percentage of (injured, infested, or damaged) bunches was 35% at Location 1, whereas this value decreased to 14.5% at Location 2.

L464- replace affected with (injured, infested, or damaged)

L467- replace “plot” with “treatment”

L520- plots (plural)

L542- leaf surface

L566- dropped to initial levels

Reviewer 2 Report

The manuscript has been significantly improved thanks to the acceptance of the numerous proposals for simplification, correction, clarification and integration of the text made by the reviewers.

Although much more balanced than the previous version, the manuscript still requires clarification in terms of "Materials and Methods" and cleanliness of the text from the numerous typing errors present especially with regard to the "References".

All my critical observations together with the correction proposal and the reporting of typos are shown below and in the pdf of the manuscript revised for the second time.

Title

Lines 4 – Replace: “releasers” with: “Releasers”

Abstract:

Line 22-23 – Replace: “alternative” with: “subsequent" or: “additional” or: “subsequent and additional”

Keywords:

Line 28-29 – In order to correctly use capital letters, and to put the different keywords in order of relevance, replace: “Pest control; European grapevine moth; Tortricidae; Field trials; passive dispensers; 28 aerosol devices; air samples; GC-MS/MS” with: “European grapevine moth; Tortricidae; passive dispensers; aerosol devices; air samples; GC-MS/MS; pest control; field trials”

  1. Introduction

Line 80 – Replace: “dispensers” with: “diffusers”

Line 130 - Replace: “alternative” with: “subsequent" or: “additional” or: “subsequent and additional”

  1. Materials and Methods

Line 140-144 - In order to make the period clearer and more readable replace: “The three different treatments included at each location were: (1) Isonet L —MD treatment with the passive dispensers Isonet® L (Shin-Etsu Chemical Co., Tokyo, Japan) at a density of 500 dispensers/ha—, (2) Checkmate —MD treatment with the aerosol devices CheckMate®Puffer® LB (Suterra LLC, Oregon, USA) at a density of 3 dispensers/ha—, and (3) Untreated Control (UTC) —without any type of treatment against EGVM—” with: “The three different treatments included at each location were: (1) Isonet L [MD treatment with the passive dispensers Isonet® L (Shin-Etsu Chemical Co., Tokyo, Japan) at a density of 500 dispensers/ha]; (2) Checkmate [MD treatment with the aerosol devices CheckMate®Puffer® LB (Suterra LLC, Oregon, USA) at a density of 3 dispensers/ha]; and (3) Untreated Control (UTC), without any type of treatment against EGVM.

Line 150-151 - Apart from the fact that on line you say that pheromone traps were checked every 14 days, can you explain which scientific publication, which logic or which common sense led you to apply the two MDT systems to the first catches of EGVM males, that, with weekly checks of the traps can have determined the application of pheromone diffusers seven days after the beginning of the first flight of the year?

To this little understandable error of approach and evaluation, it must be added that the pheromone traps after three or four weeks from their installation could have lost part of their effectiveness (attractiveness), aggravating the risk of not being able to detect the first presences of adults.

Finally, the fact that the technical data sheet of the pheromone traps manufacturer's guidelines contains the indication to replace the bait every 45 days / 6 weeks (generic prescription, entirely commercial, and not related to the foreseeable beginnings of the flights of the moth in the different wine-growing areas) is not a good reason to consider said recommendation an inviolable law, especially in the experimental field.

Having said that, I would like to invite you to reformulate these periods in a simple, adequate and convincing way, trying to overcome some obvious errors of approach.

Line 244 – Table 1 - Format and column correctly, respecting the position of the decimals!

Lines 307-309 - Neither “the population monitoring” nor “the crop damage assessments” can be considered parameters: they are activities. The actual parameters (variables) are the EGVM males captured per trap and per unit of time, and, respectively, the number of EGVM specimens (nests) per cluster (approximate to the percentage of infested bunches). Reformulate the sentence with expressions and terms appropriate to the experimental and statistical rigor.

Line 314 – Replace: “—inside the vine canopy at 1 m above the ground—“ with: “(inside the vine canopy at 1 m above the ground)

Lines 314-316 - Both the control of the traps every 14 days, and the replacement of the diffuser every 42 days (carried out regardless of a possible forecast of the beginning of the flight on the basis of the catch data of the previous years) are inadmissible choices in the experimental field.

As already previously expressed at the lines 150-151, the fact that the pheromone trap manufacturer guidelines contains the indication to replace the bait every six weeks, is not a good reason to consider this recommendation perfectly appropriate in experimental field.

Allow me to invite you to rephrase the sentence by integrating it with one or more logistical and/or cognitive justifications for the flight curves of the EGVM in the investigation areas.

  1. Results

Line 425 Table 2 - Format and column correctly, respecting the position of the decimals

Line 437 – Table 3 - Format and column correctly, respecting the position of the decimals

Lines 508-510 - Format better, eliminating dashes

References

Line 640 Format correctly according to guidelines
